# Shellfish Chitosan Potential in Wine Clarification

Veronica Vendramin [1] , Gaia Spinato [1] and Simone Vincenzi [1,2,*]

1   Centre for Research in Viticulture and Enology (CIRVE), University of Padova, Viale XXVIII Aprile 14, 31015 Conegliano, TV, Italy; veronica.vendramin@unipd.com (V.V.); gaia.spinato@studenti.unipd.it (G.S.)
2   Department of Agronomy, Food, Natural resources, Animals and Environment (DAFNAE), University of Padova, Viale dell'Università 16, 35020 Legnaro, PD, Italy
*   Correspondence: simone.vincenzi@unipd.it; Tel.: +39-0438-453711

**Abstract:** Chitosan is a chitin-derived fiber, extracted from the shellfish shells, a by-product of the fish industry, or from fungi grown in bioreactors. In oenology, it is used for the control of *Brettanomyces* spp., for the prevention of ferric, copper, and protein casse and for clarification. The International Organisation of Vine and Wine established the exclusive utilization of fungal chitosan to avoid the eventuality of allergic reactions. This work focuses on the differences between two chitosan categories, fungal and animal chitosan, characterizing several samples in terms of chitin content and degree of deacetylation. In addition, different acids were used to dissolve chitosans, and their effect on viscosity and on the efficacy in wine clarification were observed. The results demonstrated that even if fungal and animal chitosans shared similar chemical properties (deacetylation degree and chitin content), they showed different viscosity depending on their molecular weight but also on the acid used to dissolve them. A significant difference was discovered on their fining properties, as animal chitosans showed a faster and greater sedimentation compared to the fungal ones, independently from the acid used for their dissolution. This suggests that physical–chemical differences in the molecular structure occur between the two chitosan categories and that this significantly affects their technologic (oenological) properties.

**Keywords:** fungal chitosan; animal chitosan; wine clarification; dissolving acid comparison

## 1. Introduction

Chitin is the most abundant polysaccharide on earth after cellulose. Chitin is composed of 2-acetamido-2-deoxy-b-D-glucose (N-acetylglucosamine) units linked by β (1→4) bounds, and it is organized in layers of polysaccharide sheets. The sheets are composed of multiple parallel chitosan chains that could assume three different crystalline forms (α, β, γ). However, chitin is synthesized by a large number of living organisms, such as arthropods and insects (exoskeletons); crustacean (shells); and algae, plants, and fungi (cell walls) [1], mainly in its α-form, i.e., it is organized in parallel chitin chains structured in an anti-parallel sheet. In contrast, β-chitin, composed of chitin chains arranged in a parallel sheet, and γ-chitin, a mixture of the previous two forms, are quite rare. For the extraction of chitin and its derivatives at the industrial scale, two principal sources of α-chitin are suitable, such as shellfish and fungi. Annually, the seafood industry produces about 106 tons of waste [2], most of which is destined to composting or to the conversion into low-value products, namely animal feed or fertilizers [1]. As an alternative, by-products such as the shellfish shell could be directed to the component recovery, and chitosan (the deacetylated form of chitin, CTS) represents one of the best possibilities for their re-qualification. Concerning that, approximately 2000 tons of chitosan is produced every year, and its principal sources of extraction are shrimp and crab shell residues [1].

Besides, fungi represent an alternative abundant source of chitin and chitosan that could be extracted from both mycelium and spores [3]. Elsoud and El Kady [3] reported the first attempts to begin a multiple added-value compound production from fungi that

involves chitin and other compounds. It was estimated that more than 60% of the biotech industries use fungi in different processes such as brewing and baking, as well as food, antibiotics, pharmaceuticals, organic acid, and enzyme production, and that only for citric acid production, *Aspergillus niger* cultivation results in an annual waste of ~80 kton of mycelium [4]. In the choice of the source, it should be considered that the chitin structure, its percentage, and purity vary in reason of the anatomical structure in which it is located. As an example, the exoskeleton of shellfish is composed of chitin (20–30% *w/w*), proteins (20–40% *w/w*), minerals (30–60% *w/w*) [2], and by pigments and lipid in traces [5]. Insects, instead, present chitin both in the exoskeleton and in the inner parts, such as the tracheal system, that contain catecholamines –*o*-quinones allowing cross-linking between protein and chitin (36–62% *w/w* dry weight of chitin [2]). Instead, the fungal cellular wall consists of chitin (15 to 18%), β-glucans (37%), lipids (19%), and several other sugars (8 to 15%, Figure 1) [6]. However, it was demonstrated that these percentages could vary among species and life stage [7].

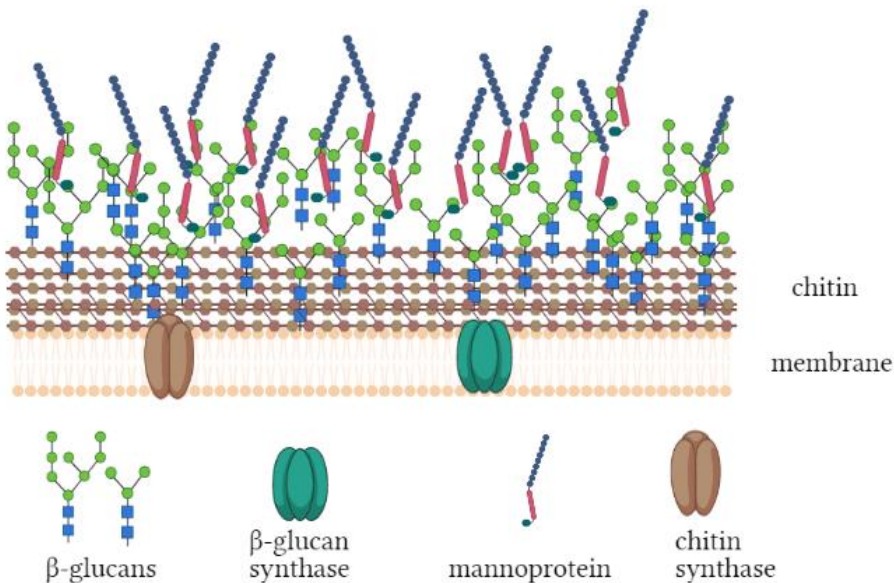

**Figure 1.** General structure of fungal cell wall.

Chitin isolation from natural material follows three steps that are different between fungal and animal CTS: the first step—which could be called "pre-treatment"—consists of the raw material washing, drying, and smashing. In the case of shellfish, in this step the minerals are removed by an acid washing, natural pigments are eliminated by means of organic or inorganic solvents, while an alkaline wash is used to remove proteins, glycoproteins, and branched polysaccharides. Instead, for the fungal chitosan extraction, an enzymatic pre-treatment of the raw materials to hydrolyze the β-glucans or, alternatively, an optimized alkaline hydrolysis at low alkali concentrations were suggested. However, Sietsma and colleagues [7] demonstrated that not all of the β-glucans composing the fungal cell wall are soluble in alkaline solution. The second step, called "deacetylation", is performed with a low amount of substrate (1:30–1:40 *w/v*) at a high alkali concentration (NaOH 1–4 M), high temperature (80–121 °C), and for short contact times (15 min–3 h) in order to remove the acetyl group from the chitin chains. The third step, called "post-treatment", generally occurs as a low concentration acid washing (frequently acetic acid at a concentration of 0.5–2% *v/v* for fungi and 2–10% *v/v* for crustaceans) that permits the recovery of deacetylated chitin (chitosan), leaving behind insoluble chitin. In fungi, the residual chitin is typically associated with the β-glucans through covalent bonds that make its recovery difficult without degradation. The amount of insoluble chitin–glucan complex could easily reach 16% of the total β-glucans [7]. After the first washing, the chitosan–acid solution is brought to pH 10 to precipitate the CTS. Finally, the precipitate is washed, commonly with

a mixture of water, ethanol, or acetone, and dried. However, several variants of this general protocol could be found in the literature [2,3] according to the producers manufacturing process. Chitosan demonstrates high plasticity, and, thus, it can be prepared in different forms, namely as films, gels, beads, and nanoparticles [1,8,9]. CTS could be used in several sectors, such as medicine, cosmetics, agriculture, and food [1,10] in light of the high number of its valuable properties, such as its biodegradability, biocompatibility, and low toxicity. CTS also exhibits high potential as antimicrobial and antioxidant agents; it could be used in the preparation of films that act as a barrier against chemical–physical changes, and the properties that it possesses by itself could be further enhanced through the combination with other useful compounds (i.e., silver, catechins, or organic acids) [10]. In winemaking, a pioneering work [11] demonstrated the possibility of using chitosan to remove phenolic compounds and increase the oxidative stability of white wines. A few years later, chitin and its derivatives were suggested to remove specific wine proteins (i.e., class IV grape chitinases) [12]. The authors found that the addition of chitin reduced the wine haze of 50% even at 1 g/L and that 20 g/L was sufficient to achieve 80% of potential haze removal. Chitosan has been admitted by the International Organisation of Vine and Wine (OIV) and European Commission since 2009 and 2011, respectively [13–15]. Since then, it spread as a fining agent for different purposes, i.e., regulation of iron and copper excess; reduction of heavy metals or possible contaminants (for example, ochratoxin); and inhibition of unwanted microbial growth, especially *Brettanomyces* spp.

Today, OIV only permits the use of fungal chitosan (from *A. niger*), in order to avoid allergenic reaction due to the crustacean material, even if the functionality and the structure of the chitosan derived from crustaceans and fungi are declared identical by the producers. Several studies tried to define the details for the optimization of chitosan extraction [4,16,17]. The most determinant chemical characters for chitosan are the deacetylation degree and the molecular weight. Previous studies discovered that the acid (organic or inorganic) used for chitosan dissolution manifests an effect on chitosan properties, such as the viscosity, mainly through the interaction with $-NH_2$ charged residue [18]. The acids used for dissolution were supposed to contribute in a different way to other chitosan properties, enhancing its antifungal activity [19] or the interaction with other compounds [20], for example. In this work, the efficiency of chitosan on wine clarification was evaluated, comparing animal and fungal chitosans. At first, an overall of 10 commercial samples were characterized for the degree of deacetylation and chitosan purity. Moreover, samples were dissolved into four different acid solutions with the aim to define whether and how this could influence viscosity and chitosan abilities in the wine fining.

## 2. Materials and Methods

### 2.1. Chemicals and Reagents

Hydrochloric acid, acetic acid, malic and succinic acid, and sodium hydroxide were purchased from Sigma-Aldrich (Milano, Italy). Water of HPLC grade was obtained by a Milli-Q system (Millipore Filter, Bedford, MA, USA).

### 2.2. Chitosan Samples

Ten chitosans (CTSs) were used for the comparison. Samples belong to two distinguished groups based on their origin, i.e., "MC" identified chitosans obtained from *Aspergillus niger* culture (samples F1, F2, F3, F4), while "SC" identified chitosan derived from shrimp shells (samples A5, A7, A8, A9) and crab shell (A6).

A more detailed description of the products is reported in Table 1.

**Table 1.** Characteristics of chitosans used for the experiments. * The MW was measured as reported below in Material and Methods.

| Sample ID | Supplier | Commercial Name/Code | Origin | Other | Calculated MW (kDa) * |
|---|---|---|---|---|---|
| F1 | Kytozime | Kiofine | *A. niger* | | 33 |
| F2 | Chibio | GBS009 | *A. niger* | High density | 84 |
| F3 | Chibio | GBS008 | *A. niger* | Chitosan oligosaccharide | 30 |
| F4 | Beijing Wisapple Biotech Co. LTD | | *A. niger* | | 49 |
| A5 | Qingdao Yunzhou Biochemistry Co. LTD | Lot. 150912 | | Food Grade (100–200 kDa) | 173 |
| A6 | Sigma-Aldrich | 48165 | Crab | Highly viscous | 478 |
| A7 | Fluka | 50494 | Shrimp | Low viscous | 51 |
| A8 | Beijing Wisapple Biotech Co. LTD | WA20170522 | Shrimp | | 282 |
| A9 | Qingdao Yunzhou Biochemistry Co. LTD | Lot. 150520-2 | Shrimp | Industry grade (100–200 kDa) | 244 |
| A10 | Qingdao Yunzhou Biochemistry Co. LTD | Lot. 150520-3 | Shrimp | Industry grade (200–300 kDa) | 228 |

### 2.3. Chitosan Deacetylation Degree

The deacetylation degree was determined by titration as described by [21], titration method I. Chitosan (0.2 g) was dissolved into 20 mL of HCl 0.1 N and 25 mL of distilled water, keeping the sample shaken at room temperature for 30 min. Then, another 25 mL of water was added, and the sample was kept at the same condition for an additional 30 min. Finally, sample solution was titrated, adding NaOH 0.1 N by an automatic titrator (Hanna Instrument, Villafranca Padovana, Italy). The degree of deacetylation (DDA) was determined by the equation:

$$DDA(\%) = 2.03 * (V_2 - V_1)/[m + 0.0042 * (V_2 - V_1)]$$

where $V_2$ and $V_1$ are volumes of NaOH corresponding to the two inflection points. Each titration curve was determined 3 times.

### 2.4. Chitosan Content

CTS content of all of the samples was determined ex novo as described by [22]. Five milligrams of chitosan powder was added to 400 µL of 10% *v/v* NaNO$_2$ and 10% *v/v* KHSO$_4$ (in the ratio of 1:1) and kept at room temperature for 15 min. After 3-Methyl-2-benzothiazolinone hydrazone (MBTH) 0.5% *m/v* addition and sample boiling, 500 µL of FeCl$_3$·6H$_2$O 0.83% *m/v* was promptly added. Samples were then cooled at room temperature, and 100 µL of each sample was transferred to a well of a 96-well microplate for the quantification at 650 nm in a microplate reader (Molecular Devices, Menlo Park, CA, USA) and expressed as the glucosamine equivalent. Data were expressed as the percentage of chitosan on effective weight, and quantification was repeated 3 times per sample.

### 2.5. Viscosity

Viscosity was chosen as a parameter to evaluate the chitosan molecular weight [23]. Analysis was performed using an Ubbelohde Viscometer type 1C (3–60 cS). Samples (1% chitosan *w/v*) were diluted 20 times in the selected buffer (acetic acid, succinic acid, malic acid, and hydrochloric acid at 1% *v/v*) before starting the measurement in order to assure that the efflux time remains below 350 s. Samples were placed in a thermostatic bath at 25 °C until thermal equilibrium, and then the time required for the efflux was measured in 2 replicates.

### 2.6. Molecular Weight Determination

The intrinsic viscosity of chitosan was determined according to the methodology of [24]. The chitosan (0.050 g) was dissolved in 100 mL of 2%HAc/0.2M NaAc, and the viscosity was measured in triplicate using an Ubbelohde glass capillary viscometer, with a viscosity range from 2.000 to 10.000 cSt (Fungilab, ASTM size 4, Sant Feliu del Llobregat, Barcelona) in a constant-temperature water bath at 25 ± 0.01 °C. The capillary diameter used was 0.63 mm. Solution concentrations were adjusted based on the viscosity of the samples, and the flow through time was kept in the range of 100–150 s. Five different concentrations were tested, and the calculation of intrinsic viscosity was obtained by common intercept of both Huggins and Kraemer plots.

### 2.7. Wine Clarification

Clarification was performed on Glera base wine furnished by Scuola di Enologia di Conegliano "G.B. Cerletti" (Conegliano, Italy), which was chosen by the results of a preliminary instability test. Chitosans were dissolved into four 1% *v/v* organic acids (malic, acetic, succinic, and hydrochloric acid) at the 1% *w/v* concentration and homogenized for 2 h by stirring at room temperature. Wine was divided into 500 mL bottles in which 5 g/hL of chitosan was added singularly to the bottles, in 3 independent technical replications. Clarification was monitored, measuring turbidity of the samples kept at room temperature (nephelometer HI 83749, Hanna Instrument, Villafranca Padovana, Italy) after 30 min, 2, 4, and 24 h after the chitosan addition, collecting 10 mL of treated wine from the bottle center.

### 2.8. Statistical Analyses

R software (R version 3.0.1) was used for statistical analysis. Differences were evaluated by a one-way ANOVA, Welch ANOVA, and Kruskal–Wallis H test depending on data distribution. The post hoc analyses Tukey HSD test and Games–Howell test were used for ANOVA and Welch ANOVA, respectively, while Dunn test with Holm correction was chosen as the Kruskal–Wallis post hoc test. Statistical significance was attributed with *p*-value < 0.05 or confidence interval of 0.95.

## 3. Results and Discussion

### 3.1. Chitosan Deacetylation

The degree of deacetylation (DDA) is a useful tool for identifying chitosan structural rigidity and its polymer conformation; in addition, it is directly connected to the chitosan (CTS) number of positive charges [23] and thus to its cross-linking attitude [25]. The high number of charged amino groups arranged on the chitosan surface facilitates its dissolution in acid solutions and guarantees a general greater functionality, i.e., the control of microorganisms, the binding of lipids, the improving of immune response, and the cytotoxic activity [26]. Nevertheless, the DDA is strongly affected by the CTS production method in light of the variation in the extraction protocols [3] that acquired even more importance when chitosans derived from different original materials are considered, as is the case of the samples studied here. Nevertheless, the identification of the original raw material cannot be sufficient to describe chitosan deacetylation and therefore, at first, the selected chitosan underwent a preliminary test that defined their deacetylation degrees.

Overall, the samples evidenced a degree of deacetylation varying between 70 and 95%, the common interval expected for commercial chitosan. CTSs could be categorized into three groups, i.e., "low" degree of deacetylation when DDA is ranging between 55 and 70%, "medium" when comprised between 70 and 85% and "high" when achieved 85–95% of DDA [26]. To date, the "ultrahigh" degree of deacetylation—DDA above 95%—is difficult to reach through an industrial process. Figure 2 shows that A9 and A10 achieved the "high" value of DDA, with 86.3% and 87.7%, respectively, while A6 evidenced the lowest level of deacetylation with 70% of DDA. The other samples were ascribed to the "medium" group. The comparison among samples highlighted a statistically significant difference between A6 and A9 and A10, with the latter grouped together ($F_{(9,19)}$ = 2.668, *p* = 0.034). It should

be noted that A6 was not completely dissolved in the buffer solution before the test, and that certainly influenced the result.

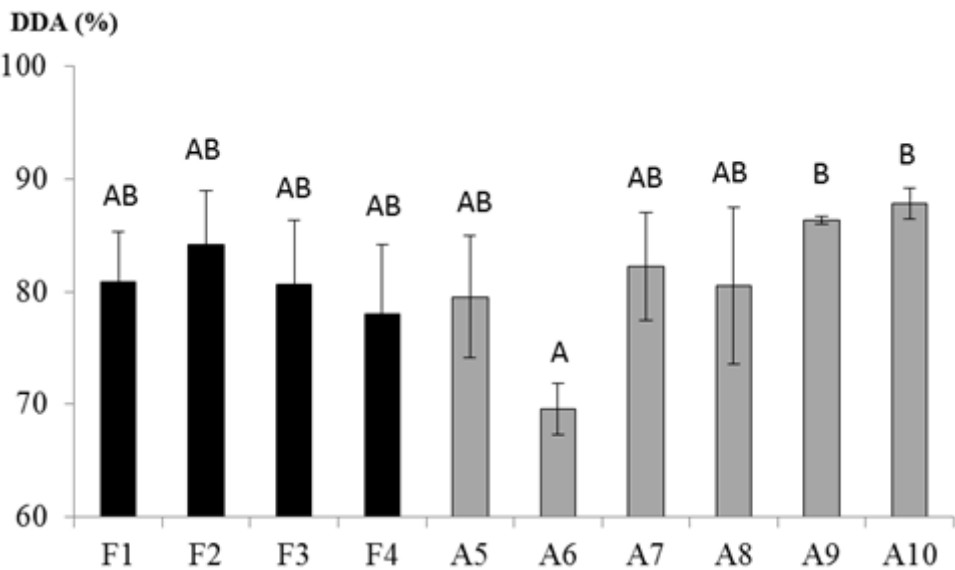

**Figure 2.** Deacetylation degree (DDA) of chitosan. Mean of three replications (in percentage) and standard deviations are expressed. Black bars: fungal-derived chitosan, light grey bars: crustacean-derived chitosan. Capital letters represent statistical groups.

*3.2. Chitosan Purity*

As previously stated, the origin of the raw material determines the chitosan physical properties. In fact, the choice of extraction protocol is based on the raw material origin and could considerably change the purity of the final extract [4]. As reported by Sietsma and colleagues [7], fungal CTS could present an insoluble percentage of β-glucan-chitin complex. Therefore, sample purity was determined by the depolymerization of chitosan into its glucosamine monomers followed by their spectrometric quantification. The amount of monomers was related to the initial sample mass, and data are expressed as percentages (Figure 3).

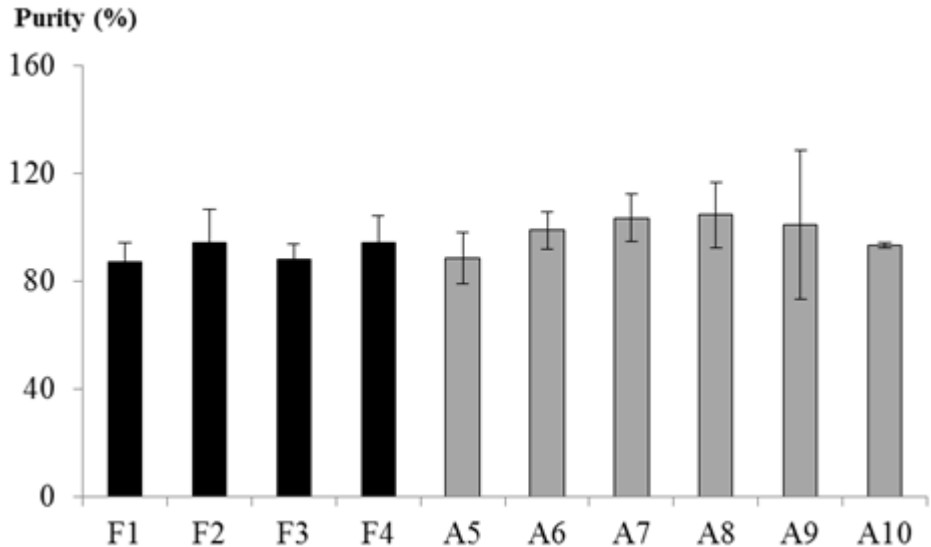

**Figure 3.** Sample purity. Mean (in percentage) and standard deviations of three replications are expressed. Black bars: fungal-derived chitosan, light grey bars: crustacean-derived chitosan.

The data show that in all of the cases the sample purity was close to 100%, with F1 and F3 as the least pure at about 87%. Statistical analyses confirmed that there was no difference among samples or between the two groups of fungal- (MC) and animal-derived (SC) chitosans.

### 3.3. Chitosan Viscosity

As is known, viscosity reflects molecular characteristics of chitosan, namely the molar mass and the surface charge [23,27]. The molecular weight was calculated as reported by [24] and is reported in Table 1. An evident lower molecular weight was registered for MC, probably depending on the enzymatic treatment necessary to reduce the glucan content on the polysaccharide extract from fungi. Regarding SC, the calculated molecular weights were generally in agreement with those, when available, declared by the suppliers, except for the sample A9. As previously mentioned, DDA also reveals a strict correlation with chitosan viscosity, in addition to the distribution of charges that could play an important role, modifying conformational behavior of chitosan. New and colleagues [28] suggested that animal and fugal chitosan could differ for CTS charge distribution. Hence, studied samples were evaluated for the viscosities expressed when dissolved in four different acids, namely acetic, malic, succinic, and hydrochloric acid. Previous experimental studies explored the effect that the dissolution acid could have on chitosan viscosity [19,29]; however, that work did not compare several chitosans or chitosans of different origin.

Statistical analyses revealed a significant difference among acid and categories, together with their interaction, as evidenced by Figure 4. The data highlight a different trend between animal and fungal chitosan upon the acid change. Moreover, shell-derived CTSs manifested higher variability than the MC, which, as expected from their low calculated molecular weights, actually did not differ in viscosity from the respective controls.

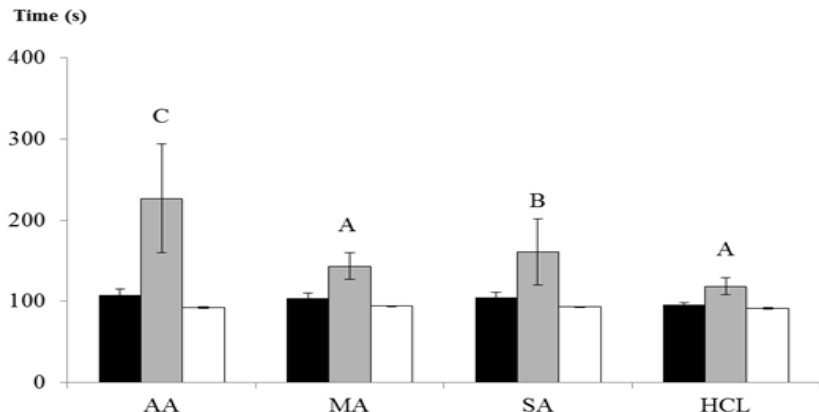

**Figure 4.** Chitosan viscosity. Mean and standard deviations (three replications for each sample) are expressed. Black bars: fungal-derived chitosan, light grey bars: animal-derived chitosan, white bars: corresponding acid solution (control). AA: acetic acid, MA: malic acid, SA: succinic acid, HCl: hydrochloric acid. Capital letters represent statistically significant differences among dissolving acids ($p < 0.05$, no letter means the absence of significance).

Figure 5 represents the time requested for the solutions to throw the glass capillary, which means that high values correspond to high viscosities. Figure 5 is focused on the animal chitosan behavior because this category evidenced the major variability. In all of the cases, CTSs revealed the highest density when dissolved in acetic acid.

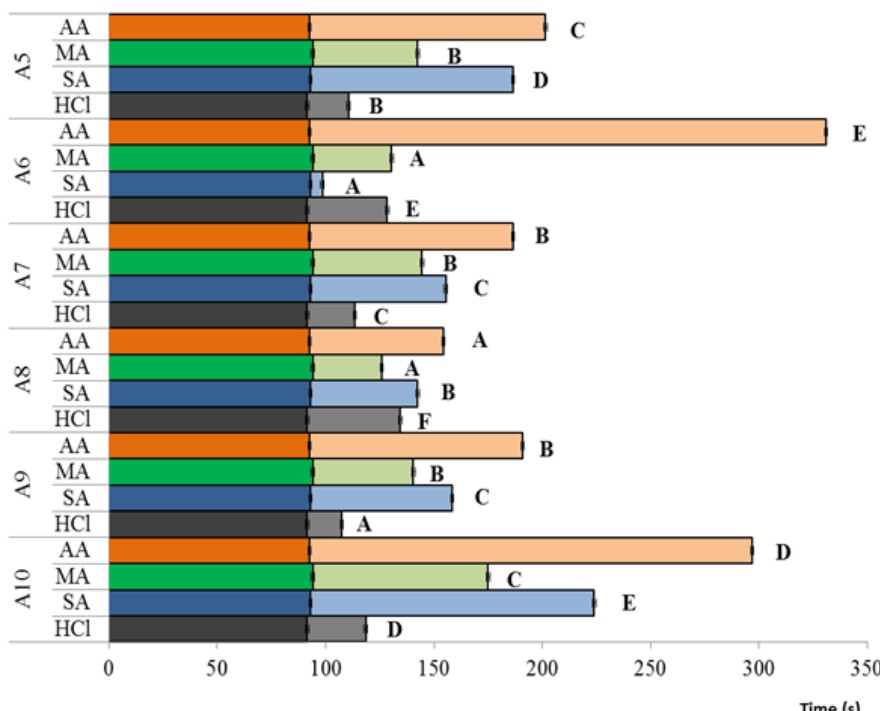

**Figure 5.** Animal-derived chitosan viscosities in four acids. Mean and standard deviations of two replications are expressed. Dark color bars: acid viscosities, light color bars: chitosan viscosities. Capital letters represent statistically significant differences among chitosans dissolved into the same acid (*p* < 0.05). AA: acetic acid, MA: malic acid, SA: succinic acid, HCl: hydrochloric acid.

Following Kasaai and colleagues [23], the constant a used for the calculation of intrinsic viscosity n is described by the equation:

$$a = [DA/(pH\ \mu)]$$

where DA is the degree of acetylation, pH is the pH, and (μ) is the solution ionic strength, demonstrating a direct relationship between the dissolution media and the chitosan viscosity. The differences recorded from the comparison of chitosan viscosities among the four dissolving acids confirmed this interaction between CTS and the dissolution system. As a matter of fact, while deacetylation could be influenced by the chitosan manufacturing as reported by Bajaj and colleagues [30], and therefore could explain the differences among chitosans, it should be assumed as constant when comparing the same chitosan sample dissolved in different acids. As explained by Kasaai et al. [23], low pH should lead to a higher degree of expansion of chitosan due to electrostatic repulsions, reducing the mobility of its structure, causing an increase in the viscosity. Unexpectedly, for each CTS, viscosities decreased following the pH lowering order (2.8, 2.6, 2.3, and 0.6 for acetic, succinic, malic, and HCl, respectively). However, it was also demonstrated that the intrinsic viscosity decreases with the increase of the ionic strength, as the chain became more flexible and compact with a reduction of the repulsive potential owing to the masking effect of anions [31]. Hydrochloric acid possesses the highest ionic strength, followed by the two diprotic acids (succinic and malic acids) and by the acetic acid. According to that, CTSs showed a reduction in viscosity when dissolved into diprotic acids and even greater when dissolved into hydrochloric acid. Moreover, Figure 5 highlights an interesting variability among chitosans in the response to acid change, which could depend on the –NH$_2$ groups available on the CTS surface. According to Cho et al. [31], the viscosity decreases because of the shielding effect of anions on the positively charged amino groups that, on one hand, induces a strong reduction of the repulsive potential, but, on the other hand, increases the

risk of flocculation and precipitation. In agreement with this, one of the studied samples (A6) showed an uncomplete dissolution in all of the acids.

### 3.4. Wine Clarification Performance

Several works explored the effects that different solvents have on chitosan properties, testing, for example, antimicrobial activities against bacteria and mold [19,32], CTS membrane properties and hydrophobicity [33], CTS film water vapor permeability [17], resistance, and elasticity [34]. However, no studies explored whether and how the choice of the acid used for the CTS dissolution influences wine clarification. Clarification is a process that occurs in nature and is linked to the flocculation and precipitation of suspended colloids, and chitosan is known to enhance this process by the instability generated by the interaction between colloids and $NH_2$ residues of chitosan [35]. Chitosan physiochemical characteristics, such as degree of deacetylation and molecular mass, affect the clarification results [36]. Studied samples evidenced heterogeneities for both DDA and molecular mass, and the viscosity test indicated that dissolution acid could affect chitosan molecular conformation. Therefore, a clarification test was performed comparing fungal- and animal-derived chitosan dissolved into the four acids. Turbidity (3593 NTU at the beginning) was recorded at 30 min, 2 h, 4 h, and 24 h. After 24 h, all of the samples demonstrated a very low turbidity, on average 112 NTU in the control, that makes the comparison difficult. Therefore, that point was excluded from further considerations.

Even though chitosan has been studied as wine fining in previous works [37,38], only the effects on the final wine composition were analyzed, without deepening its flocculation and clarification capacity. In addition, this is the first time that animal and fungal chitosans are compared in the wine clarification: the results here reported evidence that fungal CTSs efficiently remove the colloids during the treatment; however, the data showed a clear distinction between chitosan categories ($\chi^2_{(2)}$ = 49.83, p < 0.01), with fungal CTS that already reduces the wine turbidity after 30 min, by about 25%, and keeps on lowering it in the successive hours (Figure 6), and animal CTS showing a surprising clarification capacity by dropping the NTU value by about 60%. Even though the need of an acidic environment for chitosan dissolution is well known, the first chitosan-based products proposed for the enology sector were supplied as a powder to be prepared in water or directly in wine. Only recently has the market started to propose "soluble" chitosans, which already contain the acidic component needed for their dissolution. In most cases, hydrochloric acid is used (chitosan hydrochloride, CAS 70694-72-3), but other inorganic and organic acids can be used for the same scope. For this reason, the effect of the four dissolving acids, choosing among those compatible with the wine environment, on the clarification capacity was also studied. The relationship between CTS and the dissolving acid was evaluated more specifically at two time points, namely after 4 h for fungal CTS and after 2 h for animal CTS, according to the significant statistical difference detected between the two successive time points ($F_{(2, 141)}$= 15.3, p = $\leq$0.001 and $F_{(2, 213)}$ = 12.76, p = $\leq$0.001, respectively).

Figure 7 reports the turbidity values achieved from the samples 4 h after the treatment with MC. Statistical analyses revealed a significant difference among samples that is independent from the dissolving acid, expressed by the different numbers above the groups. In more detail, the sample outcomes depended—only in two out of four cases—on the interaction between the sample and the dissolving acid, namely when the higher ionic strength acids (SA, HCl) were used. These findings suggest that the variations depend on the mechanism of the primary amines protonation and probably on the CTS charge density [34]. No correlation between the calculated molecular weight and the clarification capacity was found, as the two MCs with the best clarifying capacity (F2 and F3) were those with the highest (84 kDa) and the lowest (30 kDa) molecular weight, respectively.

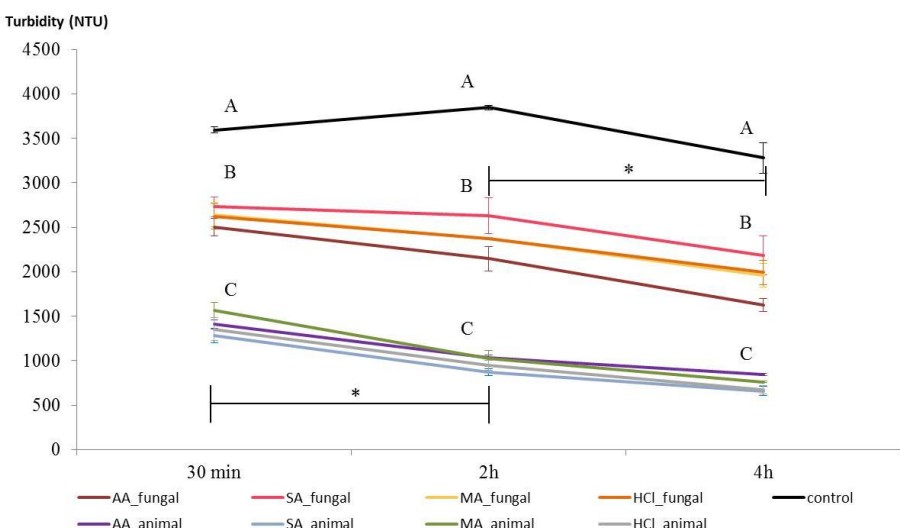

**Figure 6.** Clarification of Glera wine. Turbidities of treated and untreated wine are compared. Category mean and standard deviations of fungal and animal chitosans dissolved in four acids (three replications for each chitosan sample) are expressed. Capital letters represent statistically significant differences among dissolving acids, and stars express statistical differences between successive time points ($p < 0.05$). AA: acetic acid, MA: malic acid, SA: succinic acid, HCl: hydrochloric acid.

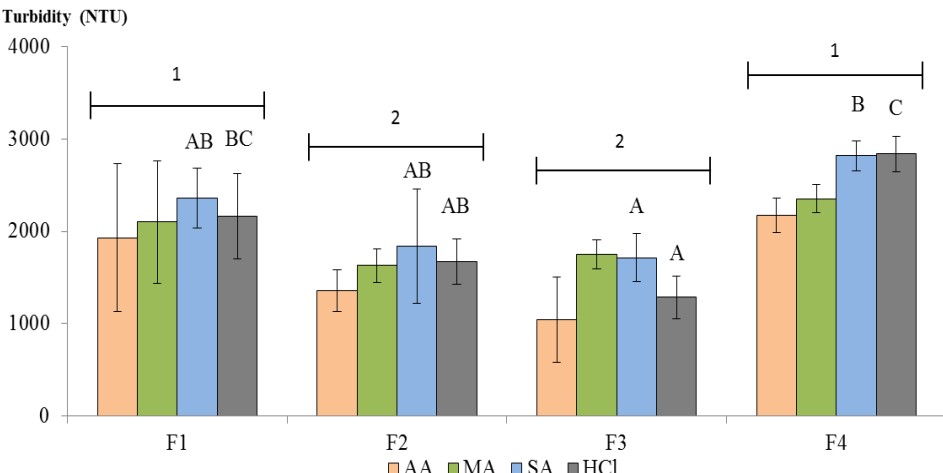

**Figure 7.** Fungal-derived chitosan clarification after 4 h. Mean and standard deviations of three replications are expressed. Capital letters represent statistically significant differences among samples dissolved into the same acid ($p < 0.05$, no letter means the absence of significance), numbers represent statistically different sample groups. AA: acetic acid, MA: malic acid, SA: succinic acid, HCl: hydrochloric acid.

Animal chitosan allowed a greater clarification than MC at all of the time points (Figure 6). Figure 8 shows the comparison of SC behavior after 2 h of treatment because after that point the chitosan clarification rate decreased. As expressed before, in animal CTS the manufacturing process varies in several steps, such as demineralization and deproteinization, besides deacetylation. Bajaj and colleagues [30] demonstrated that the alkaline deproteinization performed for 2 h could induce CTS backbone breaking even at 65 °C, or less for longer treatment, while the comparison of the deacetylations revealed a less clear effect on CTS, confirming that the "pre-treatment" participates to define CTS molecular mass. However, even in this case, no correlation between the molecular masses and the clarification capacity was evidenced. This indicates that other factors are more relevant for determining the CTS clarification property. Based on the literature, two main parameters seem to strongly affect the colloids–CTS interaction, namely the chitosan DDA

and the pH of the reaction [35]. However, A8 demonstrated a degree of deacetylation similar to A7 (Figure 2), while its clarification power was considerably lower. Concerning pH, this factor should be excluded because the experiment was carried out at the same pH value for all of the chitosans, as only one wine was used. The shellfish chitosans comparison registered a statistically significant effect of the dissolving acid (showed in Figure 8 by different capital letters), together with a significant interaction between the sample and the acid (sample x acid, $F_{(15, 48)} = 15.092$, $p < 0.01$). This is demonstrated by the fact that in five out of six cases (namely A5, A6, A7, A9, and A10) the dissolving acid had an effect on CTS clarifying capacity (expressed as lowercase letters in Figure 8), while this influence was not confirmed in the A8, which demonstrated a sensible reduction in clarification capability in comparison to the others. However, with the exception of A6 and A10, the degree of that effect was negligible. The nature of this interaction could probably be attributed to the specific charge distribution on the chitosan surface.

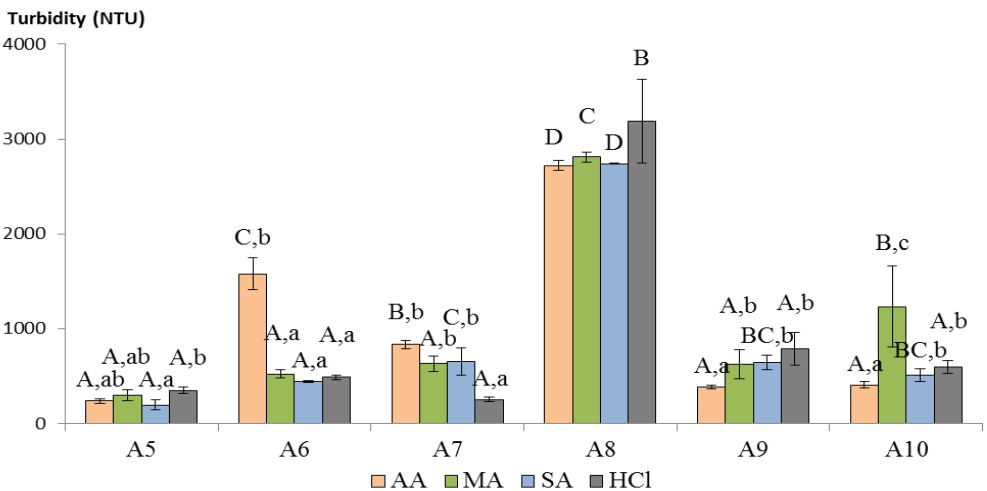

**Figure 8.** Animal-derived chitosan clarification after 2 h. Mean and standard deviations of three replications are expressed. Capital letters represent statistically significant differences among samples dissolved into the same acid ($p < 0.05$), lowercase letters represent significant differences between acids used in the sample dissolution. AA: acetic acid, MA: malic acid, SA: succinic acid, HCl: hydrochloric acid.

## 4. Conclusions

Chitosan is a natural polymer that spreads as a fining agent for microbial control, metal chelation, reduction of contaminants, and clarification in oenology. Clarification is strictly connected to the chitosan property of binding colloids, such as protein, polyphenols, polysaccharides, and metal ions. In this work, for the first time the physical effect of chitosan on clarification rate and efficiency was tested in wine. The origin of the raw material, and, consequently, the manufacturing process required for the chitosan extraction and purification, together with the efficiency in its deacetylation, are determinants in the clarification results. To date, oenological codex permits only the use of chitosan derived from fungi that, as here demonstrated, possess low efficiency in respect to the shellfish extracted chitosan. This work, for the first time, demonstrated that even under equal conditions of deacetylation and purity, the origin significantly affects clarification properties of CTS, as SC and MC are clustered separately despite the heterogeneities found within the categories. The reason of this phenomenon should be searched in the production process that probably leads to different molecular weight and charge distribution on the CTS surface. At present, no evidence of health risks in the use of animal-derived chitosan has been registered, while the recovery of useful molecules from industrial waste is generally recommended. Besides, it should be considered that chitosan from other sources, such as insect-derived chitosan, actually represents a potential source for a new generation of fining agents. Moreover, in this work, it was also evidenced that, differently from what

is registered for other applications, the dissolving acid did not significantly influence the clarification efficiency.

**Author Contributions:** Conceptualization, S.V.; methodology, G.S. and S.V.; software, V.V.; validation, G.S., V.V., and S.V.; formal analysis, G.S.; investigation, V.V.; resources, S.V.; data curation, V.V.; writing—original draft preparation, V.V.; writing—review and editing, S.V.; visualization, V.V.; supervision, S.V.; project administration, S.V.; funding acquisition, S.V. All authors have read and agreed to the published version of the manuscript.

**Funding:** This research received no external funding.

**Informed Consent Statement:** Not applicable.

**Data Availability Statement:** The data presented in this study are openly available in [repository name e.g., FigShare] at [doi], reference number [reference number].

**Conflicts of Interest:** The authors declare no conflict of interest.

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
