# Peer review of "Shellfish Chitosan Potential in Wine Clarification"

_applsci, doi:10.3390/app11104417_

Round 1

Reviewer 1 Report

The manuscript covers an interesting field and is well written. The conclusion section is interesting and highlights interesting properties of chitosan obtained from marine animals, so far not authorized for the use in winemaking. In my view the manuscript should be improved. It is not acceptable in the present form but a revised version, taking into consideration some of the suggestions and criticisms, could be considered.

Details. In the introduction, there is a rather extensive review on the production of chitosan, which could be condensed since the novelty here is limited: there have been several recent reviews on this topics published in 2020-2021.

While the title is focusing on the wine clarification, it is missing in the introduction a review of the existing evidence on the use of chitosan in winemaking. In my view this part should be expanded and major literature revised (or the title should change).

Line 144 "recently" and line 355 "quite recently". Please correct (10 and 12 years ago is not recently) 

Materials and methods. 2.2 Chitosan samples. Major criticism.The precise product code and provider MUST be added for all the commercial chitosan samples investigated. Without this information, the experiment is not replicable nor comparable with the literature and in my view should not be published.

If possible, please add also a concise paragraph with the known differences among the products tested.  At least those declared by the producers regarding the exact source from which it has been obtained. This part is optional, but would help the reader.

Materials and methods. 2.5. Wine clarification. Was the Glera a single wine sample or did they use 3 different Glera wines? If only one wine was used, please ad "technical" before "replicates".

Discussion and conclusions: please describe the novelty of the results obtained. The wine clarification trial contained in this paper is limited (which is a pity in light of the extensive characterization of the different chitosan samples) and therefore it is important to compare with previous results from the literature. Just as an example, the promising properties of chitosan from marine animals for white wine fining have been described in a pioneerin paper published in 1996 (https://www.sciencedirect.com/science/article/abs/pii/0963996996000257) 

Reviewer 2 Report

A thorough and well designed study with comprehensive chemistry analysis supported by sensory data. Valid conclusions drawn from these results.

Author Response

Non need to reply

Reviewer 3 Report

The paper by Veronica Vendramin, Gaia Spinato and Simone Vincenzi is aimed at comparing chitosan from different sources (fungi and crustacean) for wine clarification. The authors found animal and fungi chitosans to have different wine clarification behavior, despite similar structure and properties (degree of deacetylation and viscosity). This observation can be useful, but the authors need to understand the reasons for this behavior deeply. The paper looks incomplete at this point, and I recommend rejection. Some additional experiments need to be performed to identify the reasons for the observed behavior of chitosan samples extracted from various sources.

Specific comments:

  • To identify the reasons for the different behavior of animal and fungi chitosans, it is necessary not only to analyze the chitosan content but also to identify impurities (proteins, glucans, minerals, etc.), which will be different for animal and fungal chitosan samples and impact their properties.
  • Figure 2: The degree of deacetylation was determined with a large inaccuracy (often more than +-5%). This leads to statistically insignificant differences in this parameter between samples. I recommend the authors use an additional method (recommended by the US and EU pharmacopeia) - 1H NMR spectroscopy. In addition to a more accurate determination of the degree of deacetylation, NMR spectroscopy will help evaluate the presence of organic impurities in the samples.
  • Even though the authors conducted a comprehensive viscometric study; however, the average viscosity molecular weights are not given in the paper. The European Chitin Society does not recommend the publication of the research papers on chitosan, in which three main characteristics (source, molecular weight, and degree of deacetylation) are missing.
  • An essential characteristic of chitosan samples is the molecular weight distribution, which can also affect their properties, including wine clarification. I strongly recommend authors perform size-exclusion chromatography (e.g., SEC-MALS) to compare the molecular weight distribution (dispersity) of the chitosan samples.

Reviewer 4 Report

The manuscript “Shellfish chitosan potential in wine clarification” presents very interesting results. However, the experiments were only conducted over a single wine, which presented low turbidity 24 h after starting the experiment and without chitosan addition. In my opinion the authors should perform additional experiments using a different wine, in order to guarantee that the results obtained are independent of the wine used.

Line 141: The origin of sample A10 is missing.

Line 229: Replace “loss” by “less”.

Lines 246-247: Figure 4 only presents statistical analysis among dissolving acids for SC chitosans, so the phrase should be reformulated.

Figure 6: The turbidity at 0 min. and at 24 h is important to be indicated.

Figure 7: Why are only presented the statistical results for succinic and hydrochloric acid?

Lines 323-325: The phrase should be reformulated as the statistical effect of the dissolving acid is not presented.

Line 326-327: The statistical data to support this statement is not presented, please reformulate.

Lines 348-352: These two sentences are quite confusing. The analysis of the results should be clarified.

Conclusions: Should present more conclusions from the present work.

Round 2

Reviewer 1 Report

The revised version of this manuscript has been significantly improved and in my view it can be published

Author Response

see asee Attachment

Reviewer 3 Report

Despite the fact that the authors did not take most of my comments into account, the article as it stands meets the minimum requirements for publication.
